# Information Gathering about Pregnancy, Birth, and Puerperium—Good and Fake Information

**DOI:** 10.3390/ijerph20064848

**Published:** 2023-03-09

**Authors:** Oezden Oyman, Joy Fest, Roland Zimmermann, Nicole Ochsenbein-Kölble, Ladina Vonzun

**Affiliations:** 1Faculty of Medicine, University of Zurich, 8006 Zurich, Switzerland; 2Department of Obstetrics and Gynecology, Baden Regional Hospital, 5404 Baden, Switzerland; 3Department of Obstetrics, University Hospital of Zurich, 8091 Zurich, Switzerland

**Keywords:** information, pregnancy, Internet, health professionals

## Abstract

Recent research on the subject of information-gathering processes among pregnant women has revealed a shift towards online sources. Health professionals’ knowledge about sources of information has been shown to improve the understanding and counseling of patients. The objective of this study was to create an overview of all types of sources relevant to information gathering and to put their role and perception into perspective. Methods: A total of 249 women were included in this study and recruited over a period of one month at the University Hospital of Zurich (USZ). Exclusion criteria included cases of fetal demise and late abortions. The survey on information-gathering processes was divided into three stages: pregnancy, birth, and puerperium. The different sources of information were compared based on women’s characteristics. Results: The response rate was 78% (n = 197). The main findings include a significant difference in information gathering based on varying levels of education, with women at the lowest educational level using the Internet the least during pregnancy (*p* = 0.029). During puerperium, significant differences could be observed in the involvement of the gynecologist. Primipara women as well as women of lower educational levels contacted their gynecologist less in contrast to multipara women (*p* = 0.006) and women of higher educational levels (*p* = 0.011). Overall, health professionals were considered to be the most important source of information. Conclusions: This study demonstrates that parity and educational level influence the information-gathering process. As the most important source for information gathering, health professionals must use this advantage to better assist their patients in accessing reliable information.

## 1. Introduction

Currently, information gathering about pregnancy, birth, and puerperium in Switzerland is limitless. With the arrival of the digital age, there are fast and easy ways to obtain information. The Internet has a major impact on the information-gathering process and is frequently used and appreciated by most pregnant women [1,2,3,4]. Furthermore, access to health professionals is comparably simple, and high-quality brochures and leaflets are often distributed by healthcare providers. Nevertheless, pregnant women tend to pre-inform themselves and conduct further research about the information disclosed by their doctors, either to feel more confident about received information or because they felt underinformed [2,3]. It has been shown that health professionals’ knowledge about different sources of information gathering among their patients improves understanding and counseling. Previous research has primarily focused on how pregnant women currently gather information online and how the reliability of the retrieved information is assessed [1,2,3,4,5,6,7,8]. In contrast, this study focuses on the perceived performance of accessible information.

The aim of this study was to create an overview of various types of sources relevant to information gathering concerning pregnancy, birth, and puerperium and to assess the role and perception of such sources. Ultimately, the objective was to better understand the current situation and the relevance of different sources in order to help health professionals adapt their approach in supporting pregnant women.

## 2. Methods

This prospective survey was conducted in 2018 over a period of one month at the University Hospital of Zurich (USZ). All women who delivered and were receiving care in the hospital’s postpartum ward were asked to participate. Women who had experienced fetal demise or late abortion were excluded. Each woman was approached and asked to complete the survey in person. The checklist was filled out privately and collected within two days. The time needed to answer the questions was estimated at approximately 15 min. The study does not fall within the scope of the Human Research Act of Switzerland and therefore did not need approval by the local ethics committee (BASEC Nr. Req-2019-00428).

### 2.1. Checklist

Checklists were available in German or English, either in paper format or online using SurveyMonkey.com, a common survey platform. A total of 36 questions regarding demographics, previous pregnancies, and information gathering were queried. In addition to the multiple-choice format, there was also space provided for open answers. Questions on information gathering were addressed separately for pre-conception, pregnancy, birth, and puerperium stages and included a rating of the perceived usefulness of the different sources during each phase. Information topics were general and not specifically restricted to medicine. Furthermore, respondents were asked to evaluate pregnancy-related websites, to depict how well informed they felt on a scale from 1 to 100, and if they missed particular information, wished they had taken a different approach to information gathering, or were subjectively falsely informed by any source. The content of the misinformation was not queried.

### 2.2. Statistical Analysis

The different sources of information were compared by women’s characteristics (where, for education, “no degree” includes the completion of primary school, secondary school, and/or gymnasium). Cases in which these characteristics were not stated were excluded from analyses to avoid bias. Data were analyzed using Excel and Statistical Package for Social Sciences (SPSS, Version 26). Cross-tables and Pearson’s chi-square test were used where appropriate to compare categorical variables. A *p*-value of <0.05 was considered statistically significant.

## 3. Results

During the study period, 249 births satisfying the inclusion criteria were registered. Of those, 197 (78%) women could be included in the study. Dropouts comprised the following: 22 women were not found in their rooms, 14 could not participate due to a lack of language skills, 12 chose not to participate, and 4 did not return the checklist. The demographic characteristics of the respondents are presented in Table 1.

### 3.1. Pre-Conception

Sixty-six percent of the respondents (131 out of 197) informed themselves before pregnancy. As sources were identical to those used during the pregnancy, details are summarized in the following section.

### 3.2. Pregnancy

Sources of information and their perceived usefulness regarding pregnancy are shown in Table 2.

The sources most used by respondents were primarily the gynecologist, followed by family and friends and, lastly, the Internet. Their respective perceptions as “very useful” are as follows: gynecologist, 70%; family and friends, 38%; and Internet, 39%. In unexpected or otherwise special circumstances, women first consulted a gynecologist (38%, 74 out of 197 participants), the Internet (28%, 55 out of 197 participants), or family and friends (18%, 35 out of 197 participants). A significant difference in Internet use between women of different educational levels was also observed (*p* = 0.029). A total of 71% of the respondents without a degree (20 out of 28 participants) used the Internet, whereas 88% of the women with tertiary education (116 out of 132 participants) and 94% of the women who completed a traineeship (34 out of 36 participants) gathered information online.

### 3.3. Birth

Sources of information and their perceived usefulness regarding birth are shown in Table 3.

The sources most used by respondents were predominantly family and friends, followed by the gynecologist and, lastly, the Internet. Their respective perceptions as “very useful” are as follows: family and friends, 34%; gynecologist, 61%; and Internet, 32%.

There are significant differences in the sources for information gathering between women with varying educational levels, namely among cohorts addressing family and friends (*p* = 0.035), the gynecologist (*p* = 0.026), and the midwife (*p* = 0.027). A total of 97% of women who completed a traineeship (35 out of 36 participants) consulted their family and friends, whereas 80% of women with tertiary education (105 out of 132 participants) and 75% of the women without a degree (21 out of 28 participants) referred to them. A total of 94% of women who completed a traineeship (34 out of 36 participants) consulted the gynecologist with open questions, whereas 78% of the women with tertiary education (103 out of 132 participants) and 68% of the women without a degree (19 out of 28 participants) consulted similarly. A total of 75% of women who completed a traineeship (27 out of 36 participants) involved the midwife, whereas 60% of the women with tertiary education (79 out of 132 participants) and 50% of the women without a degree (14 out of 28 participants) did the same.

### 3.4. Puerperium

Sources of information and their perceived usefulness regarding puerperium are shown in Table 4.

The sources most used by respondents were, in order of frequency, family and friends, the midwife, and the gynecologist. Their respective perceptions as “very useful” are as follows: family and friends, 31%; midwife, 73%; and gynecologist, 28%.

There are significant differences in the sources for information gathering between women with varying educational levels, namely in addressing the Internet (*p* = 0.042), the gynecologist (*p* = 0.011), and the midwife (*p* = 0.001). A total of 43% of women without a degree (12 out of 28 participants) used the Internet, whereas 65% of women with tertiary education (86 out of 132 participants) and 72% of women who completed a traineeship (26 out of 36 participants) consulted similarly. A total of 43% of women without a degree (12 out of 28 participants) contacted a gynecologist, whereas 65% of women with tertiary education (86 out of 132 participants) and 81% of women who completed a traineeship (29 out of 36 participants) did so. A total of 46% of women without a degree (13 out of 28 participants) referred to a midwife, whereas 75% of women with tertiary education (99 out of 132 participants) and 86% of women who had completed a traineeship (31 out of 36 participants) did the same. A significant difference in using the gynecologist between women of different parities was also observed (*p* = 0.006). A total of 56% of primiparas (65 out of 115 participants) consulted the gynecologist, whereas 78% of multiparas (61 out of 79 participants) did the same.

### 3.5. Websites

A total of 140 out of the 197 respondents (71%) consulted at least one of the websites listed in Table 5. Swissmom.ch was the most commonly used (52%) and most appreciated (57%) website.

### 3.6. Misinformation, Rating, and Future Pregnancies

Of the 197 respondents, 48 (24%) stated that they felt misinformed at least once during pregnancy. Misinformation was mainly derived from the Internet or friends (40% and 25%, respectively). Overall, the mean rating value of the level of information was 79 on a scale from 1 to 100. For future pregnancies, 44 (22%) participants would not change anything in their approach to information gathering, whereas 119 (60%) reported that they would consult a specific source more frequently in the future. Of these women, 23% stated that they would involve the midwife and 18% stated that they would involve the gynecologist more often.

## 4. Discussion

This study provides a comprehensive overview of the sources of information gathering about pregnancy, birth, and puerperium and shows that patients’ characteristics, in particular parity and educational level, clearly influence this process. A number of aspects deserve further evaluation.

### 4.1. Sources of Information Consulted by Respondents

Health professionals: The first and most important source (based on overall usage and usefulness during the corresponding stages) are health professionals, namely the gynecologist and midwife. Routine prenatal appointments, as integrated into the Swiss healthcare system, enable participating health professionals to provide information at different phases. A Dutch study on women’s experience with information sources during pregnancy found that more than 80% of women found professional information sources trustworthy and useful [9].

Nevertheless, in our study, 60% of the women were not satisfied with their individual approach to information gathering, and the majority stated that they would prefer to involve the gynecologist and midwife even more in their information gathering for future pregnancies. These findings emphasize even more so the important role of healthcare providers.

Internet: Eighty-seven percent of respondents used the Internet as a source of information. These results are supported by previous findings demonstrating that the majority of women (as many as 94%) refer to the Internet for information [1,2]. Moreover, our results are in line with the previously mentioned Dutch study, where nearly 80% used websites as a common information source [9]. The frequency of Internet usage as well as its perceived usefulness decreased for the stages of birth and puerperium (from 87% to 74% and 63%, respectively). This could be explained by the involvement of the midwife, who can provide more practical knowledge.

Former experience: There is a significant difference worth mentioning based on gravidity and parity. Multigravidas and -paras tend to more frequently refer to the gynecologist and midwife, while other sources are not perceived as very useful. It is likely that their past experiences influence this behavior and, moreover, that they depend less on other individuals and their perceived experiences (family, friends, Internet, etc.). These women may have also missed some information on a certain stage in their previous pregnancy and opted to seek more specific and professional advice. In contrast, primigravidas and -paras more frequently felt that these external experiences and reports proved helpful. This difference is most evident for consulting the gynecologist during puerperium.

Educational background: Women without a degree (primary and/or secondary education) proceed quite differently in gathering information compared to women with a higher educational level. Women without a degree were less likely to consult the Internet for information about pregnancy and puerperium and were also less likely to consult the midwife and gynecologist for information concerning the upcoming birth and puerperium. Possible explanations for these differences are potential cultural reservations, language barriers, and/or a lack of financial resources among women without a degree. It is also possible that well-read women with higher educational levels are more likely to consult every available source. The Dutch study suggests that women with lower levels of education were more likely to prefer text-limited sources, using visual images next to plain language, to receive their health information [9]. Clearly, this consideration should not be underestimated when preparing pamphlets and webpages. Despite the varying uses of sources, the perception of their usefulness was independent of educational background. Nevertheless, it is important to mention that only a few participants (15%) compose the category “no degree” and that some respondents did not even rate their information sources, making these data particularly prone to errors. In order to verify these findings, a study with more participants would be required. The Internet was mostly consulted by respondents who had completed a traineeship. However, previous studies demonstrated that women with tertiary education use the Internet most often [1,6,7]. This may be explained by variations in educational systems and could be further explored in future research.

### 4.2. Clinical Relevance

In this study, it was observed that respondents were generally satisfied with the information they received (79 out of 100 points). However, given the seemingly endless amounts of input available on the Internet, women should be mindful of false information. Nearly a quarter of women in this study were misinformed at least once, mostly by the Internet and/or friends, and experienced dissatisfaction with the information available. Although concerning, these results are at the same time reassuring when compared to previous studies showing that more than half of the women read misleading or wrong information online and wished for recommendations for websites from their doctors [2,3]. In any case, this subject remains a major challenge, as, in the coming years, the demographics will shift more towards the “digital natives”, who are accustomed to social media and are more likely to be exposed to unreliable sources. Moreover, they might follow social media influencers and ultimately be more vulnerable to false information. To avoid such misinformation and confusion provided by the Internet, professionals should act as scientific and clinically experienced influencers, promote reliable websites, and, furthermore, support their patients with data retrieval, interpretation, and application [3,5,8]. Thus, clearly, behavior on the Internet must be better understood by health professionals [10]. During consultations, health professionals should allow for timeslots to address this issue. Another approach could be to involve the midwife at an earlier prenatal stage before patients are admitted to the delivery room, as demonstrated by examples from Australia [11] and Sweden [5]. In these studies, the midwife was consulted by most of the women, as their maternity care systems integrate midwifery care throughout pregnancy. Moreover, it could be shown that women felt more confident in asking a trained person for relevant information or entrusting them with their concerns. A further valued approach is the distribution of information booklets by health professionals, as seen in Australia [11]. While this survey was conducted in 2018, the earlier pamphlets published by the USZ were replaced with revised handbooks [12]. They now serve health professionals as an initial guide and basis in supporting high-quality information gathering for patients.

### 4.3. Limitations

The main limitation of this study is its design as a single-center cross-sectional study based at the USZ. Therefore, the findings may not be applicable to other patient populations. In order to evaluate nationwide trends, this survey could be expanded to other hospitals and clinics. Further improvements could include expanding demographic data (for example, including marital or employment status) and adding more specific questions (such as details concerning the frequency of using sources) to allow for the evaluation of specifying questions about detailed information gathering.

## 5. Conclusions

This study provides a comprehensive overview of sources for information gathering related to pregnancy, birth, and puerperium and shows that patients’ characteristics including parity and educational level clearly influence this process and perceptions. Despite the shift of usage towards the Internet, health professionals are still considered the most important source for information gathering. These understandings should encourage health professionals to adapt their approach in supporting information gathering for pregnant women and also encourage them to embrace their role as scientific influencers.

## Figures and Tables

**Table 1 ijerph-20-04848-t001:** Demographic characteristics of the respondents.

N = 197	n (%)
Age	
<30	55 (28)
31–35	71 (36)
>36	71 (36)
Nationality	
Swiss	83 (42)
Foreign	114 (58)

**Table 2 ijerph-20-04848-t002:** (a) Consulted sources and their usefulness regarding the pregnancy, distinguished by number of pregnancies. (b) Consulted sources and their usefulness regarding the pregnancy, distinguished by educational level. Significant results are marked bold.

(a)				
n (%)	Primigravida		Multigravida	*p*-Value
**N = 195**	97 (50)		98 (50)	
Family and Friends	85 (88)		85 (87)	0.361
Very Useful	33 (39)		24 (28)	
Gynecologist	85 (88)		89 (91)	0.182
Very Useful	56 (66)		66 (74)	
Midwife	57 (59)		57 (58)	0.843
Very Useful	32 (56)		29 (51)	
General Practitioner	34 (35)		37 (38)	0.912
Very Useful	12 (35)		9 (24)	
USZ Website	54 (56)		50 (51)	0.778
Very Useful	11 (20)		8 (16)	
USZ Pamphlet	52 (54)		50 (51)	0.934
Very Useful	16 (31)		13 (26)	
USZ Courses	34 (35)		31 (32)	0.719
Very Useful	14 (41)		6 (19)	
Internet	84 (87)		85 (87)	0.828
Very Useful	38 (45)		28 (33)	
Books and Articles	73 (75)		65 (66)	0.208
Very Useful	22 (30)		11 (17)	
**(b)**				
**n (%)**	**No Degree**	**Traineeship**	**Tertiary Education**	***p*-Value**
**N = 196**	28 (14)	36 (19)	132 (67)	
Family and Friends	21 (75)	32 (89)	119 (90)	0.121
Very Useful	12 (57)	18 (56)	36 (30)	
Gynecologist	24 (86)	33 (92)	119 (90)	0.742
Very Useful	16 (67)	24 (73)	83 (70)	
Midwife	14 (50)	22 (61)	79 (60)	0.857
Very Useful	9 (64)	16 (73)	36 (46)	
General Practitioner	15 (54)	13 (36)	44 (33)	0.315
Very Useful	8 (53)	5 (14)	8 (18)	
USZ Website	13 (46)	21 (58)	72 (55)	0.915
Very Useful	2 (15)	6 (29)	11 (15)	
USZ Pamphlet	12 (43)	22 (61)	70 (53)	0.690
Very Useful	4 (33)	6 (27)	19 (27)	
USZ Courses	13 (46)	11 (31)	43 (33)	0.617
Very Useful	4 (31)	3 (27)	13 (30)	
Internet	20 (71)	34 (94)	116 (88)	**0.029**
Very Useful	8 (40)	15 (44)	43 (37)	
Books and Articles	16 (57)	24 (67)	100 (76)	0.238
Very Useful	4 (25)	5 (21)	24 (24)	

**Table 3 ijerph-20-04848-t003:** (a) Consulted sources and their usefulness regarding birth, distinguished by number of births. (b) Consulted sources and their usefulness regarding birth, distinguished by educational level. Significant results are marked bold.

(a)				
n (%)	Primipara		Multipara	*p*-Value
**N = 194**	115 (59)		79 (41)	
Family and Friends	93 (81)		66 (84)	0.073
Very Useful	35 (38)		14 (21)	
Gynecologist	87 (76)		67 (85)	0.062
Very Useful	46 (53)		48 (72)	
Midwife	71 (62)		48 (51)	0.328
Very Useful	42 (59)		20 (40)	
General Practitioner	31 (27)		28 (25)	0.378
Very Useful	8 (26)		5 (18)	
USZ Website	50 (43)		36 (46)	0.457
Very Useful	11 (22)		4 (11)	
USZ Pamphlet	58 (50)		39 (49)	0.571
Very Useful	17 (29)		6 (15)	
USZ Courses	37 (32)		19 (24)	0.146
Very Useful	10 (27)		2 (11)	
Internet	84 (73)		59 (75)	0.699
Very Useful	28 (33)		18 (31)	
Books and Articles	68 (59)		46 (58)	0.369
Very Useful	20 (34)		6 (13)	
**(b)**				
**n (%)**	**No Degree**	**Traineeship**	**Tertiary Education**	***p*-Value**
**N = 196**	28 (14)	36 (19)	132 (67)	
Family and Friends	21 (75)	35 (97)	105 (80)	**0.035**
Very Useful	8 (38)	9 (26)	37 (35)	
Gynecologist	19 (68)	34 (94)	103 (78)	**0.026**
Very Useful	9 (47)	24 (71)	62 (60)	
Midwife	14 (50)	27 (75)	79 (60)	**0.027**
Very Useful	9 (64)	17 (63)	36 (46)	
General Practitioner	10 (36)	17 (47)	33 (25)	0.075
Very Useful	4 (40)	4 (24)	5 (15)	
USZ Website	11 (39)	21 (58)	56 (42)	0.293
Very Useful	1 (9)	5 (24)	9 (16)	
USZ Pamphlet	11 (39)	23 (64)	65 (49)	0.170
Very Useful	2 (18)	7 (30)	14 (22)	
USZ Courses	9 (32)	14 (39)	35 (27)	0.328
Very Useful	3 (33)	4 (29)	5 (14)	
Internet	18 (64)	30 (83)	97 (73)	0.475
Very Useful	5 (28)	10 (33)	31 (32)	
Books and Articles	14 (50)	24 (67)	78 (59)	0.198
Very Useful	3 (21)	5 (21)	19 (24)	

**Table 4 ijerph-20-04848-t004:** (a) Consulted sources and their usefulness regarding the puerperium distinguished by number of births. (b) Consulted sources and their usefulness regarding the puerperium distinguished by educational level. Significant results are marked bold.

(a)				
n (%)	Primipara		Multipara	*p*-Value
**N = 194**	115 (59)		79 (41)	
Family and Friends	90 (78)		62 (78)	0.146
Very Useful	25 (28)		14 (23)	
Gynecologist	65 (56)		62 (78)	**0.006**
Very Useful	32 (49)		24 (39)	
Midwife	83 (72)		58 (73)	0.163
Very Useful	63 (76)		41 (71)	
General Practitioner	25 (22)		28 (35)	0.206
Very Useful	9 (36)		6 (21)	
USZ Website	48 (42)		30 (38)	0.217
Very Useful	11 (23)		3 (10)	
USZ Pamphlet	60 (52)		47 (59)	0.402
Very Useful	23 (38)		10 (21)	
USZ Courses	41 (36)		26 (33)	0.397
Very Useful	10 (24)		5 (19)	
Internet	69 (60)		53 (67)	0.142
Very Useful	17 (25)		10 (19)	
Books and Articles	57 (50)		42 (53)	0.408
Very Useful	15 (26)		4 (10)	
**(b)**				
**n (%)**	**No Degree**	**Traineeship**	**Tertiary Education**	***p*-Value**
**N = 196**	28 (14)	36 (19)	132 (67)	
Family and Friends	18 (64)	32 (89)	103 (78)	0.065
Very Useful	10 (56)	9 (28)	28 (27)	
Gynecologist	12 (43)	29 (81)	86 (65)	**0.011**
Very Useful	6 (50)	12 (41)	16 (19)	
Midwife	13 (46)	31 (86)	99 (75)	**0.001**
Very Useful	11 (85)	23 (74)	71 (72)	
General Practitioner	7 (25)	15 (42)	29 (22)	0.219
Very Useful	3 (43)	3 (20)	9 (31)	
USZ Website	8 (29)	18 (50)	53 (40)	0.412
Very Useful	1 (13)	4 (22)	9 (17)	
USZ Pamphlet	11 (39)	23 (64)	74 (56)	0.231
Very Useful	5 (45)	4 (17)	24 (32)	
USZ Courses	9 (32)	15 (42)	43 (33)	0.717
Very Useful	3 (33)	3 (20)	9 (21)	
Internet	12 (43)	26 (72)	86 (65)	**0.042**
Very Useful	5 (42)	6 (23)	16 (19)	
Books and Articles	9 (32)	20 (56)	71 (54)	0.132
Very Useful	2 (22)	2 (10)	15 (21)	

**Table 5 ijerph-20-04848-t005:** Websites and their usefulness.

N = 197	Used, n (%)	Very Useful, n (%)
Swissmom.ch	102 (52)	58 (57)
Famigros.migros.ch	33 (17)	5 (15)
Wireltern.ch	28 (14)	7 (25)
Windeln.ch	31 (16)	3 (9)
Buggyfit.ch	14 (7)	1 (7)
Website of the USZ	60 (30)	24 (40)
Website of other hospitals	24 (12)	9 (38)
Others	49 (25)	19 (39)

## Data Availability

Data are available in the main text.

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
