# Peer review of "Information Gathering about Pregnancy, Birth, and Puerperium—Good and Fake Information"

_ijerph, 2023, doi:10.3390/ijerph20064848_

Round 1

Reviewer 1 Report

This article is written clearly and with a correct structure, but in my opinion, the study period was too short (one month), it should be extended over a longer period. The results of the study are predictable. The study is too old (2018), in 5 years the information may change.
Line 14...correct month, not moth.
Line 262...please correct the end of the sentence.
You have used few references (8 references). They should also be updated with newer ones (the last reference is from 2020).
If the article will be accepted for publication, I recommend you to add the questions from the questionnaire as supplement.
Even if it is an anonymous study, the patients' informed consent is also required and should be approved by an ethics committee.
Author Contributions must also be completed.

Author Response

This article is written clearly and with a correct structure, but in my opinion, the study period was too short (one month), it should be extended over a longer period. The results of the study are predictable. The study is too old (2018), in 5 years the information may change.

We thank you for your encouraging introducing remarks. We chose the study period too be one month as we think an N=249 to be a good and representative number. Of course, we are aware of the fact, that presenting higher numbers is always better. However, by choosing the period of a month we could ensure a participation rate of nearly 80%, which is extremely high for an enquiry and in the sum resulting in very solid study results.

We also agree that in times of digitalization changes in sources of information may change rapidly. And, we agree that the websites might have slightly changed since or that channels like i.e. tik tok were not considered in this study. This cannot be changed retrospectively. A strength of our center, however is, that digitalization was pushed very early and that the pamphlets/App and Webpage were already available for all of our patients and did not significantly change as source of information.

Line 14...correct month, not moth.

Thank you for pointing out this spelling mistake, which was corrected accordingly

Line 262...please correct the end of the sentence.

The sentence was corrected accordingly.

You have used few references (8 references). They should also be updated with newer ones (the last reference is from 2020).

Interestingly not that much literature is found on the specific topic; however, we managed to provide an update of the reference list.

Even if it is an anonymous study, the patients' informed consent is also required and should be approved by an ethics committee.

This is a very important point, which we were critical about as well. We therefore did apply for an ethical approval and received the following answer:

‘The project you are planning does not fall within the scope of the Human Research Act and therefore does not require the approval of the Cantonal Ethics Committee for its implementation.’ BASEC-Nr. Req-2019-00428.

The request number was added into the text accordingly and the wording was adapted.

Line 58-60:  The study does not fall within the scope of the Human Research Act of Switzerland and therefore did not need approval of the local ethics committee (BASEC Nr. Req-2019-00428).

Author Contributions must also be completed.

Thanks. The author’s contributions were completed.

Reviewer 2 Report

Dear authors

This an interesting survey about Information gathering about pregnancy, birth, and puerperium. Howeverthere are some points that I think will help improve the article.

The above comments are attached.

Author Response

Reviewer 2:

Key words:

  1. It is better to choose the keywords from MeSH, so that they can be found more easily in the search

Thank you for pointing out this relevant issue. We adapted the keywords accordingly

Line 28-29: Keywords: Information; Pregnancy; Internet; Health professionals;

Methods:

  1. I am not sure that this research does not require the approval of the Ethics Committee, because any research with any method requires the informed consent of the individuals and considerations of ethical issues, and these must be approved by the Ethics Commettee.

This is a very important point, which we were critical about as well. We therefore did apply for an ethical approval and received the following answer:

‘The project you are planning does not fall within the scope of the Human Research Act and therefore does not require the approval of the Cantonal Ethics Committee for its implementation.’ BASEC-Nr. Req-2019-00428.

The request number was added into the text accordingly and the wording was adapted.

Line 58-60:  The study does not fall within the scope of the Human Research Act of Switzerland and therefore did not need approval of the local ethics committee (BASEC Nr. Req-2019-00428).

  1. Give more explanation about the questionnaires. How are they developed? How about their reliability? Have they been validated? If yes, mention in the method part; and if not, use the term of checklist instead of questionnaire.

This is a relevant precision, thank you! We adapted the wording accordingly and now use the term ‘checklist’ throughout the text.

Results:

  1. It is recommended to state the results of the study on all the questions at once (pre-conception, pregnancy, birth and post-partum) and then expressed in the form of subgroups separately, as mentioned in the discussion section

Tables are all repetitions of text. The data mentioned in the tables should be removed from the text or vice versa.

We thank you for this suggestion. After careful consideration and discussion, we prefer to keep the presentation of the results in the current order. We agree, that text and tables have a certain redundancy. However, the tables serve to provide a complete summary of our data, deleting information does not seem sensible to us. Ultimately, we are in conflict regarding the shortening of the text, since the journal rather advises us to extend the text.

If the editor prefers some adjustments, we are of course ready to do so.

  1. Define ‘Misinformation’. What does it mean?

‘Misinformation’ is a synonym of false or incorrect information.

Discussion:

  1. In the discussion section, the results of the study should be mentioned only if needed, and there is no need to mention the p-value.

We removed all p-values and unnecessary results from the discussion.

Limitations:

  1. This study is a cross-sectional type and tot a cohort study.

We adapted designation of the study type accordingly.

Round 2

Reviewer 1 Report

The authors completed all recommendations. Congratulations for your work!

Author Response

Thank you very much for your valuable review comments!

Reviewer 2 Report

Dear author, this is an interesting survey aboutInformation gathering about pregnancy, birth, and puerperium” with Remarkable results.

1.  It is routine for articles that the tables be as descriptive as possible to avoid repeating them in the text as much as possible. However, if the policy of the journal is in its current form and the respected editor accepts it, there is no problem.

ï¼’.  I know the meaning of the word “misinformation”. My suggestion was to make a clear definition of misinformation regarding the patients in this study. What do patients consider as false information and what were their criteria for it? Was it just their own opinion or was a definition already assumed for it?

Author Response

Please excuse the misunderstanding to your input of the 'misinformation'. The checkliste allowed us to identify misinformation as a subjective perception only. We asked for the source of it but did not ask for the content. Retrospectively, this does not allow us to check for its correctness.

We addded a little precision into the methods part.

Line 70-74: 

Furthermore, respondents were asked to evaluate pregnancy-related websites, to depict how well informed they felt on a scale from 1 to 100, if they missed particular information, wished they had taken a different approach to information gathering, or were subjectively falsely informed by any source. The content of the misinformation was not queried.